# Addressing Complex Social Problems Using the Lens of Family Violence: Valuable Learning from the First Year of an Interdisciplinary Community of Practice

**DOI:** 10.3390/ijerph20043501

**Published:** 2023-02-16

**Authors:** Heath Greville, William House, Stella Tarrant, Sandra C. Thompson

**Affiliations:** 1Western Australian Centre for Rural Health, University of Western Australia, Geraldton, WA 6530, Australia; 2Law School, University of Western Australia, Perth, WA 6009, Australia

**Keywords:** community of practice, value creation, interdisciplinary, complex problems, teaching and learning, violence against women, family and domestic violence

## Abstract

It is imperative that universities continue to explore innovations that support staff and student learning and pursue their mission to promote social responsibility and community service. Communities of Practice have been used to facilitate innovation and regenerate teaching and learning in tertiary contexts, including interdisciplinary collaborations around complex problems. This study describes the challenges and achievements of the first year of an interdisciplinary Community of Practice which aimed to create innovative approaches to teaching and learning about family and domestic violence, a complex social issue, inherently gendered, which receives little attention across the University discipline areas, despite the centrality of this issue in much of the future work of University graduates within a range of professional areas. We interviewed engaged members to explore the value gained from their first year of involvement in the Community of Practice. This initiative brought members substantial value while recognising the need for long-term engagement and commitment from the senior University leadership to embed innovation. A key lesson was that developing an innovative curriculum to address critical and ongoing social and public health issues requires much more senior leadership, responsibilities shared across faculty, and commitment of dedicated resources and staff time. The findings provide valuable learning for other Communities of Practice attempting to engage with complex problems and create innovative interdisciplinary approaches to teaching, learning, and research.

## 1. Introduction

In the complex world of the 21st century, many major challenges cannot be solved by the simple application of science and technology. Solutions to global problems require interdisciplinary thinking and collaboration. As always, existing knowledge and practice will be an important basis from which new approaches and knowledge emerge. Although universities are rooted in historical traditions, they must examine their approaches both to the content they cover and to the pedagogy of building new knowledge.

Communities of Practice are receiving increasing attention in both the health and education sectors as an approach to fostering learning and sharing information in the workplace. This paper explores the application of a Community of Practice to a multidisciplinary learning approach which had, at its heart, efforts to educate students and faculty to examine, reflect on and think differently about the complex social problem of family and domestic violence. We first describe the concept of a Community of Practice (CoP), then provide background on the complex social issue of family and domestic violence (FDV) and the relevance of a CoP to reforming ways in which students are trained to think about more effectively addressing the multi-layered problems related to FDV. The challenges experienced and the value created for members and stakeholders during the first year of the CoP are discussed.

### 1.1. Communities of Practice

Communities of Practice, a phrase first coined by Wenger, is an approach to organisational learning, with the definition evolving over the years. As of 2011, Wenger and colleagues define a community of practice as a “learning partnership among people who find it useful to learn from and with each other about a particular domain. They use each other’s experience of practice as a learning resource” [1]. Wenger calls this form of group communication “social learning”, which is “the exchange, acquisition, and evaluation of knowledge that occurs in the participative context of a group of practitioners”. Wenger sees the benefit of this type of learning in creating a holistic understanding of an area of knowledge for each individual member of the community as they share knowledge and fill in gaps that may arise for other members. The informal nature of a CoP means that membership is voluntary, and the loose leadership structure [2] allows horizontal relationships to form, thus organically preserving the community. Discussion leads to new ideas and projects, which lead to new discussions, and all members can contribute to the collective building of knowledge which is the “social learning” [3]. De La Rue adds that technology can aid in dealing with geographical distances, relevant to the example we describe. De La Rue also recognises limitations in horizontal relationships and the connections that form between members resulting from physical separation [2].

At its core, there are three key elements that must be met for a group to be able to claim itself as a “community of practice” [4]. These key elements were articulated by Wenger but have since been expanded on by other researchers such as Smith and colleagues [5] and Pharo et al. [6]. The first element is the Domain; the focus of knowledge that leads to the CoP being formed, with a collective understanding of that focus. Secondly is the Community, the group of individuals involved, and this is achieved through the interactions, relationships, and social norms and values that are shared amongst the group finally, the Practice, represents the compilation of work and approaches that the CoP has formed to discuss the “domain”.

The literature describes an agreed structure of a community of practice as having three “rings”, separating the three types of members [2,3,7]. The core members are the facilitators, leaders, and activators, all contributing regularly and planning the meetings and events of the community. Within a multi-disciplinary CoP, the facilitator has a particularly important role, as they act as the bridge between all the disciplines, aiding in the communication between varying schools of thought and in the formation of ideas, events, and products [6]. Without the facilitator, collaboration would be lacking, projects would not grow past the conceptual stage, and continuity and record-keeping of the CoP could not be achieved. Outside this core group are the active users, who contribute to the conversation to a lesser extent than the core group and require a direction from the core group [2]. Finally, the peripheral users are those who rarely contribute, instead listening to the active users and absorbing the knowledge. The ability of a member of a CoP to be in the “ring” that they feel most comfortable with and to move between “rings” as workloads change, allows the CoP to be inclusive, and coupled with the passion for the domain of knowledge that is required, a CoP is usually quite diverse [6]. However, unless a member is in the core group, they rarely have a formal role in the CoP and so commitment and collaboration can be limited with their everyday work roles being prioritised over the work of the CoP. The size of the CoP is also as important as the structure of the group, with a critical mass being required for an effective discussion to be had [4,6]. Smaller groups do not have the critical mass to have discussions, nor do they have the breadth of varying knowledge to benefit from a structure such as the CoP. A larger group can therefore allow effective discussions, but also runs the risk of being too large, limiting both continuity of members and participation [4].

Interdisciplinary learning “draw[s] together and integrate[s] different forms of knowledge to explore a problem and produce insights that are more than the summing of disciplinary parts [6]. Research highlights the obvious result of each discipline working in isolation from the other, in that each discipline is looking at the same issue from different “siloed” views, failing to see the full scope of the topic [8]. This leads to each discipline failing to make an impact in responding to the issue [8,9]. Pharo and colleagues found that the various university disciplines are radically different in terms of teaching methods, content taught, and organisational culture, and students who took one or two units from disciplines different from their major faced inconsistent and even contradictory learning experiences. In contrast, students who took several units from varying disciplines were able to manage the conflicting nature of the units, as well as being more able to respond to social issues that require multiple ways of thinking [6].

The University of Western Australia (UWA) has an explicit commitment to cross-disciplinary work “promoting more just societies that foster equitable allocation and enforcement of rights for all their members, particularly those most marginalised by historical and institutional processes” (University of Western Australia) and in December 2018, the University invited staff to form multidisciplinary CoPs for innovation and best practice in higher education teaching and learning. A small amount of funding was available to selected applicants with constraints around how funding could be used and a 12-month timeframe for spending it. There was keen interest in working with university colleagues to submit an application from staff working in a university-administered department of rural health based in Geraldton, a regional city 420 km north of Perth. There, within the City of Greater Geraldton, collaborative efforts were occurring to implement an FDV primary prevention strategy called the Community Respect and Equality (CRE) Plan, through support for businesses and organisations to address attitudes, behaviours, and norms that promote family violence [10]. The UWA staff based in Geraldton were committed to interprofessional student learning and could see the benefits of bringing academic expertise from outside of health sciences to support the efforts of the CRE and better responses to family violence. This prompted several academic and general staff to develop a CoP using the lens of Family and Domestic Violence (FDV) to explore complex social problems to promote interdisciplinary student learning and social justice. Social justice refers to justice in terms of the distribution of wealth, opportunities, and privileges within a society. Principles of social justice include access, equity, participation, and human rights, all relevant to addressing FDV. For simplicity, this community of practice is hereafter referred to as the FDV CoP.

### 1.2. Complex Social Issues and Family and Domestic Violence FDV

Complex social issues are severely complex problems that resist definition and for which scientific, planned, rational responses are insufficient [11], with solutions requiring a different mindset [12]. Debate surrounding the place of higher education in society has highlighted the responsibility of universities to educate students to respond to complex issues [9,13]. Researchers in the area of complex problems and communities of practice have pointed out that addressing complicated problems from within a disciplinary framework is itself a complex problem: there is the need to transcend disciplinary boundaries [6] and to achieve both depths in understanding and breadth in the range of complex issues that must be considered [14,15].

In the CoP, we used the term Family and Domestic Violence although Australia’s national policy response explicitly identifies the gendered nature of this issue and uses the term violence against women and their children. However, in Western Australia, the state government uses the term Family and Domestic Violence in their 10-year strategy and so that was the term we adopted for the CoP, and in this paper. FDV is defined as “the threat or exercise of physical, psychological, and/or emotional violence, i.e., any type of force against another person with the intent of inflicting harm or exercising power and control over them”, with the perpetrator being an individual in a close relationship with the victim, not necessarily living with them. The impacts of FDV on a person victimised by it can be serious and long-lasting, affecting an individual’s health, well-being, education, relationships, and housing outcomes [16]. FDV is rarely a single physical attack but rather a complex system of abuse that is physical, verbal, emotional, financial, and mental, and it extends far beyond physical harm to the victim. Impacts of FDV on individual victims include depression, addiction, anxiety, and post-traumatic stress disorder. The complex and pervasive effects of FDV make it a public health issue, requiring holistic approaches to prevention and response [17]. Within the CoP, the gendered nature of FDV was discussed and understood. We did not focus on sexual abuse as often it is physical violence that elicits a victim reporting or law and justice responses, and we were keen to broaden awareness of the pernicious nature of coercive control which is under-recognised, more difficult to deal with, and where public responses are underdeveloped. 

In Western Australia (WA), rates of family violence are higher in rural areas. Living in rural and remote areas may limit a victim’s ability to leave a violent relationship and their ability to access formal and informal support. Fear of stigma, shame, and the community view that violence is a “family problem” further deter victims from disclosing their experiences of abuse [16]. The City of Greater Geraldton, a regional local government area located 420 km north of Perth with a population of 38,000, has FDV rates more than twice the state average [18].

### 1.3. University Teaching and Learning about FDV

Inevitably, given the complex causes and wide-ranging impacts of FDV, university teaching, learning, and research related to FDV potentially occur across multiple disciplines, based within various Faculties and Schools including Law, Media, Public Health, Social Work, Education, Medicine, and Public Policy. Our CoP included members from all the above disciplinary areas except Education. One CoP member reviewed teaching about FDV within the Faculty of Health and Medical Sciences and found that across the faculty at the undergraduate level, there was just one lecture on FDV in an elective unit. At post graduate level, Social Work students had one three-hour block and there were a few “mentions” of FDV across faculty teaching, for example in Medicine, within the context of alcohol use. FDV was used as an example within some Public Health lectures, where staff had a particular interest in the area but apart from these small, isolated offerings, there was no structured study of this complex social issue. Furthermore, teaching focused on responses to violence through professional roles (e.g., social work, medicine) and did not explore underlying causes of family violence. Within the Law School, a Gender Studies unit offered teaching and learning about the causes of FDV as situated within gender inequality, but the CoP members knew of no other FDV-specific teaching in the university. 

The professionals produced by the university (social workers, lawyers, policy writers, teachers, etc.) usually work within organisations offering one or two discrete services. Women and children experiencing FDV and seeking assistance with their complex situation may find themself accessing a wide variety of organisations and professional services which are not well integrated, may use different jargon, and may assume different causes of the problem and prioritise different outcomes for the person. Consequently, the web of professional services accessed by women and children experiencing FDV demonstrates the complexity, as was mapped by a member of the CoP.

Each of the shapes in the diagram indicates a service staffed by individuals, many of whom have had a university education but who likely learnt virtually nothing during their tertiary studies about (1) the causes of FDV, (2) the sheer complexity of services to be negotiated by the victim, or (3) the professional framework, point of view or terminology of the other service providers and professionals in the picture. Moreover, the extent to which any lens regarding gender and its social dimensions was captured in any teaching regarding FDV was unknown but considered likely to be minimal.

The FDV CoP recognised that limited pre-service education about FDV delivered within a curriculum that treats the issue in fragmented disciplinary silos, along with the lack of coordinated government services results in confusion and frustration for people experiencing violence. This complexity of issues related to fragmented FDV services and professional approaches (Figure 1) requires shared understanding in teaching and learning along with transdisciplinary approaches which integrate academic and non-academic modes of knowing [14]. The university recognised in its New Century Campaign that a “safe, equitable, and progressive world depends on educating today’s youth to be tomorrow’s global citizens” (URL https://giving.uwa.edu.au/prioritiest, accessed on 14 January 2023). The CoP aimed to bring together academic and professional staff with experience and interest in innovation, service learning, complex problems, and social responsibility, and to enrich teaching and learning by focusing on Family and Domestic Violence (FDV) to explore interdisciplinary and innovative approaches to complex problems and policy advocacy. By bringing the experiences of practitioners and service users into teaching, learning, and research, disciplinary silos can be broken down or at least made more open to other perspectives and understandings. The CoP proposed interdisciplinary expert involvement towards better teaching and learning, ultimately improving services for those impacted by FDV and strengthening efforts to prevent FDV.

An evaluation of the effectiveness of the first 12 months of the CoP was undertaken, including the use of Wenger and colleagues’ 2011 conceptual framework for assessing the value created by CoPs [1]. The framework proposes five cycles of “value creation” that a CoP may engage in multiple times during its life, with different cycles producing different kinds of value: (1) Immediate value (activity/interactions that have value in and of themselves); (2) Potential value (human capital, social capital, tangible capital, and learning capital); (3) Applied value (changes in practice); (4) Realised value (performance improvement) and (5) Reframing value (fundamental changes in understanding how success is defined, including strategies, goals, and values). More detail of the framework is described in Appendix A.

## 2. Methods

To describe the processes and effects of the FDV CoP, we undertook a qualitative documentary analysis of the CoP’s output (meeting minutes and event reports) and an analysis of interviews with 12 members who had attended at least half of the eight CoP meetings in the 12 months from January 2019 to December 2019. The intention to undertake this evaluation was discussed with CoP members in meetings as the Educational Enhancement Unit which provided funding for the CoP encouraged evaluation of the initiative. 

Questions were developed by the evaluation team and included information on their position at the university, involvement in the CoP and what it meant for them, and what they saw as the challenges and achievements of the CoP, including their suggestions going forward with respect to continuing the CoP. Interviews were set up in advance at a time convenient for the staff member and conducted individually, voice-recorded, and transcribed. Each interviewee was sent their transcript by email and invited to make corrections as needed to ensure accuracy. The evaluation was undertaken with the most regular attendees at the CoP meetings and represented those based in law (2), and health, albeit from very different disciplinary backgrounds including public health/health promotion, medicine, rural health, research, and social sciences, (9), and media and communications (1). 

Analysis was undertaken in two stages. Each interview was separately analysed for benefits, challenges, and suggestions for ways forward. Through multiple rereading of interview transcripts, debriefing, and discussion among the authors, an agreement was reached on the findings. Each interviewee received their transcript and was invited to elaborate on points to enable validation of meaning. Secondly, two of the evaluation team also used deductive coding on all interviews to analyse each one based on keywords in Wenger and colleagues five levels of value. The email correspondence, meeting notes, and reports of activities of the CoP were also re-examined to apply the same framework of analysis to assess the value created. A draft of the findings from the evaluation was circulated to all CoP members providing an opportunity for their feedback.

## 3. Results

### 3.1. Participants, Process Measures of the CoP, and Its Development over Time

The initial CoP expression of interest and discussion involved 26 academic and non-academic staff and students, and one practitioner from an external agency. Members came from three Faculties, seven Schools, the Public Policy Institute, Student Services, and one external organisation. Three core members were from the School of Population and Global Health within the Faculty of Health and Medical Science and the Law School within the Faculty of Arts, Business, Law, and Education. As with all CoPs, over the course of a year, membership fluctuated, with new members joining and others dropping back to accommodate the other professional demands on their time.

Aims and objectives of the CoP:

The first meeting of the CoP proposed several potential areas of focus, aligned with the possibilities in the original expression of interest, namely to:Collect information to broaden understanding of the issues and current responses,Create teaching and learning resources,Aim for policy impact: from primary prevention through to tertiary services,Raise awareness of and improve services on campus and links to possibilities outside of campus,Engage in collaborative research on FDV.

In the first meeting, CoP members discussed the potential of transformative learning and a paper on the subject [19] sparked discussions about what could be achieved within the CoP. One member with extensive experience in service learning and syllabus design developed a “mind map” which made evident that creating a teaching module that would do justice to the complexity of FDV was beyond the scope of the CoP given the limited funding available and time constraints. 

While some members were output-focused, others wanted more opportunities to learn about the work of colleagues in different disciplines. At its second meeting, the CoP agreed to hold a symposium to bring members from outside the University, create transdisciplinary connections, enable deeper exploration of the issues from multiple points of view, and provide opportunities for future collaborations.

The multidisciplinary symposium titled “How can we Think, Act, and Communicate Differently to improve the capacity of graduates and staff to facilitate and accelerate innovative work and research in Family and Domestic Violence prevention and response?” (May 2019) was held about one-third of the way through the academic year. It was an all-day event split into a series of presentations in the morning followed by an “open space” event with free discussion on topics of interest. Forty-four individuals attended the symposium, including representatives of 13 external organisations, CoP members, and students from the university who had either studied a unit or completed a professional practicum related to FDV. The symposium inevitably surfaced the gendered nature of FDV and achieved the beginning of a transdisciplinary perspective and working relationships with agencies, bringing “real world” perspectives and collaboration opportunities. Key outcomes included plans to develop community-based student educational experiences focused on FDV prevention in a regional/rural setting and progressing FDV as a UWA Grand Challenge, an initiative that the university had floated as a way of harnessing cross-disciplinary action and innovation to solve important national or global problems.

In addition to bringing together diverse individuals with an interest in family violence, the symposium showcased numerous research projects that were underway in the FDV space, previously unknown to many attendees. In the words of one member, this allowed the CoP to “realise that actually there’s a critical mass of [FDV] researchers in the university”, raising the possibility of increasing the awareness of policymakers external to the university about the potential of UWA to contribute to this important area.

Following the symposium, in the second half of the year, three cross-disciplinary teaching and learning initiatives involving “real world” learning developed directly from connections made through the CoP. Two collaborative research opportunities were pursued and a well-attended end-of-year event which included student presentations on FDV-related research confirmed the value and potential of the CoP to continue to engage with students and staff in interdisciplinary projects.

### 3.2. Findings from the Qualitative Analysis

The interviews highlighted several recurring themes regarding CoPs as a model in general, reiterating findings from the broader literature. Four distinct themes were identified: motivation for participation in the CoP, achievements, challenges, and wishes for the future of the CoP.

#### 3.2.1. Motivation for Participation in the CoP

Ten of the twelve interviewees joined the CoP due to their passion and interest to make a difference in tackling issues related to family and domestic violence, with some members explicitly saying that they wanted to fill gaps in their own knowledge and learn from other disciplines. One member was motivated by the opportunity to give their students’ industry experience, and one respondent noted their motivation was primarily an interest in interdisciplinary research. This supports Wenger’s proposition that an effective CoP requires a domain of shared interests and passions in the area [5].

Respondents reported different levels of interdisciplinary and cross-campus connection prior to joining the CoP. Five members felt that they had connections with disciplines across the university, but four felt that these connections were with similar disciplines to their own and that their connections with different schools and faculties were almost non-existent. Most Geraldton-based members reported no interdisciplinary connections at the university, a consequence of operating from a small rural campus, but these members found themselves more closely connected to the wider community locally, including non-university organisations and industry members.

Several participants commented on the limited effectiveness of university functions to create cross-disciplinary relationships, one member describing such formal functions as simply “fleeting interactions”. However, three members with pre-existing interests in interdisciplinary approaches to learning were members of the UWA Medical Humanities Network, a committee that “straddled disciplines” and which functioned as a community of practice that provided an opportunity for educators and researchers engaged in medicine, allied health sciences and dentistry and those in the humanities, arts, music, psychology and social sciences to exchange ideas, share information and opportunities, and establish collaborative partnerships. 

#### 3.2.2. Achievements

Three areas of achievement of the CoP were identified: cross-disciplinary collaboration; increased knowledge and awareness of the university; and the development of connections outside of the university.

Cross-disciplinary collaboration

Overwhelmingly participants noted that the largest achievement of the CoP was that “it brought together a lot of different people from different disciplines with an interest in family violence”. This not only allowed for a holistic discussion on FDV, but also for the gaps in knowledge to be realised, and for the member’s own knowledge of FDV to be expanded. Interdisciplinary relationships were able to form, opening future possibilities for cross-disciplinary research. Some members expressed a strong desire to work with specific colleagues, and some disciplines were interested in working further with other disciplines.

Bringing together individuals with an interest in family violence and not only the CoP, but it also became evident that a large amount of research that was being completed in the FDV space, previously unknown to other members. The CoP allowed members to discover a critical mass of researchers interested in FDV “in close proximity”, increasing the awareness of policymakers inside and outside the university about the potential of the university to contribute to this important area.

One member commented that the core idea of the CoP, the interdisciplinary approach, “is what is key, interdisciplinary but also intersectoral collaboration to address the issue, so it isn’t siloed into one area... it’s a whole of society approach if any kind of cultural change is achieved”. This comment regarding the CoP’s achievement in promoting the interdisciplinary approach addresses criticisms in the broader literature regarding the failure of contemporary research methods to prevent the siloing of knowledge [8,9]. Several members commented on the CoP as an effective forum for raising ideas that could lead to future projects, and although such projects had not yet come to fruition, the ideas and networks that were created showed the success of the CoP as a forum for discussion.

Increased knowledge and awareness of the university

Another perceived achievement of the CoP was increased knowledge and awareness of the university overall, as members became more aware of university structures as well as smaller projects. The members from the main (Nedlands) campus initially had limited or no understanding of work happening in Geraldton, but over time they became very aware of the rural work and the possibility of linking projects there. Increased connections with student placements occurred, including media students doing work alongside industry partners in Geraldton regarding perceptions of FDV. Connections were also established with the McCusker Centre for Citizenship, which had an existing unit in Wicked Problems. Their focus on Wicked Problems had up to that time been urban-focused although it aligned well with the CoP’s aim to explore complex social problems and foreshadowed a future collaboration to develop a cross disciplinary focus on FDV through student placements in Geraldton.

Development of connections outside of the university

The third generally identified achievement of the CoP was the development of connections outside of the university in the form of growing relationships with industry members. This included external organisations, sites hosting student placements, and other student-centred organisations. The Symposium was seen as the main representation of these developing connections, as a variety of organisations and individuals presented and discussed their own FDV-related work and conversed with others around their organisational needs and possibilities.

Alongside addressing the siloing of knowledge, the CoP discussed possibilities for creating curriculum material that was holistic and would aid graduates in developing a broader understanding of their future work. A benefit of the CoP was to promote holistic learning, “mirroring what we want our students to do upon graduating”, which required teachers to reflect on how to prepare their students to work with complex social issues. The symposium and discussions with industry members gave an opportunity for both academics and external organisations to be grounded in the wider world; to discuss what is happening outside the university and to hear about industry needs including the skills and knowledge that were valued in graduates to support them being effective in their jobs. The symposium and the conversations with non-university stakeholders also had the benefit of increasing opportunities for future external collaborations, “encouraging each other to look outward from the university so that our work is not just university-based”. Therefore, the CoP was seen as not only creating a holistic understanding of an issue or area of society but also as grounding such knowledge in real-world applications, breaking down the silos of different disciplines, and engaging the university more broadly in its aim of preparing graduates for the real world and developing better analysis and responses to its complex demands.

#### 3.2.3. Challenges

Managing time

By far, the largest challenge reported in engaging with the CoP was time management. Conflicts in scheduling often resulted in CoP involvement being pushed aside. Some members were away during the year and so missed meetings due to external reasons, but many simply had conflicts in timetabling with teaching or administrative demands and needed to prioritise their work over their attendance at CoP meetings. As one member commented, the CoP:
*…sat as something they could only give time to once they had done all the other things they needed to do. There wasn’t a sense of university roles and expectations, it wasn’t an outcome you could say ‘I’m putting in time into this and I’m going to get a paper out or I’m going to publish it so I can put it down for my teaching’.*

So, in terms of personal benefits it seems more akin to something benefiting the community and the university as opposed to something that will help in your career.

This sentiment was shared by others who prioritised other work demands over CoP involvement. The members who attended all meetings were those with formal roles within the CoP who felt an obligation to attend and participate but also had opportunities to influence the scheduling of the meetings. Again, these comments mirror the literature which has noted that universities rarely value this added collaboration external to academics’ core roles within the university; hence, unless there is a personal benefit to involving themselves regularly in the CoP, even passion for aiding an improved response to the issue of FDV is diminished by deadlines, workloads, and other commitments [9]. However, if projects and research papers arose from the CoP activities, it could give an added incentive to participate in the CoP, especially working collaboratively with the other disciplines.

Short timeframes and limits on the use of the funding

Regarding the creation of projects, many found that constraints of funding and the timeframe made the possibility of implementing tangible projects a difficult task. The funding initiative that supported the FDV CoP proscribed paying for dedicated staff or student time, and the original timeframe was for 12 months. Many participants raised this as an issue, as with a complex problem such as family and domestic violence, it takes months of discussion to reach a common understanding on which tangible projects could be based. A much longer timeframe was considered necessary for many proposed projects to be realised. As this participant described
*There’s a lot of developmental work happening in the first year or 6 months and relationship building …So as it matured there would be more opportunity for more tasks and formal outcomes.*

The key facilitator’s role was regarded as critical and underpinned the effectiveness of the CoP, with many compliments on her abilities and commitment to sharing information (such as videos and articles) and organising meetings and events. Without this key facilitator role, it was acknowledged that very little would have been achieved. 

Balancing openness to new members with achieving outcomes

Linked with the workload and valuing issues discussed previously was the problem of continuity of membership at meetings. The inclusive nature of the CoP meant that new members could join at any time, and with each new member, meetings tended to reiterate discussions to enable an attendee to be brought up to speed on not only the current topics and events of the CoP but also the foundational knowledge of the issue of FDV and each ’member’s different perspective. This meant some regular attendees felt meetings lacked adequate focus on outcomes and weakened the ability of the CoP to be efficient. Inconsistent attendance led to the perception by some that the CoP was “going around in circles”, as one member put it, in a constant round of introductions and exploration of possibilities, without settling on a tangible product to work on over the ever-contracting remainder of the year. 


*...last time we met we had new people come and we covered the same ground that we covered in previous meetings-because every time there’s a new person, we talk about what we might do.*


Although the CoP was inclusive, a perceived weakness was limited involvement from several key areas of the university where understanding of FDV was important. Those in Education, preparing teachers to work with children affected by FDV, were conspicuously absent. Several discussions were held in the CoP regarding how to get more men involved in the conversation, and this challenge also was raised by four interviewees. The CoP aimed to promote collaborations and regularly discussed the potential to create educational material to use in educating their respective students. School teachers were regarded as important stakeholders who have contact with children affected by FDV but are often inadequately prepared to deal with the trauma children experience because of that exposure to FDV. This lack of involvement in Education was seen to leave a gap in the interdisciplinary FDV discussion. The challenge of engaging men, whilst not detrimental to the interdisciplinary aspect of the CoP, was nevertheless seen to weaken the CoP’s deliberations. Robust discussion and leadership by men about their role within society, and directly in addressing FDV was regarded as critical, and academic leadership by men was regarded as part of this. Lack of connections to the university’s broader decision-making structures at an organisational level was felt as a weakness of the CoP’s ability to achieve university-wide change.

Balancing exploration and creative input with structure and project development

From the first meeting, there was a tension between the desire for tangible products such as teaching resources, and the desire to deepen understanding through wide exploration of the issues, to “think about our thinking”, and to explore the social conditions and epistemologies that give rise to FDV. Members were also keen to bring to light the views of students, practitioners, and external agencies on university teaching about FDV. All these ideas were subject to the pressures of short time frames, with tension over the urgency of producing tangible outputs. Two members commented that they would have liked more time to explore issues. Examining the causes and impacts of FDV involves deep exploration into the fabric of gender relations, social norms, and potentially the foundations of teaching practice. A decision to dive into planning projects risked applying fragmented and limited understandings of the issues into these endeavours. In the words of one CoP member:
*Do you immediately try to do something about the problem that you see now, or do you look underneath it and see what are the things that are causing the problem?*

Creating and sustaining new ways of working

Three members commented on the challenges of truly achieving cross-disciplinary perspectives and new ways of working together.


*The biggest challenge was trying to think creatively about the problem of the CoP, how do we really work together? How do we really break down the discipline silos and how do we really create ways of crossing them? How do we change the Law [Units] and how do we change the way we work together, it’s very hard…*


Another issue raised was the challenge of the physical separation of CoP members based at the rural campus. While videoconferencing was used for meetings, technological issues sometimes prevented members from feeling part of the discussion, and the time and cost of travel as well as competing demands impeded regular in-person meetings, with most campus-based members not having the opportunity to visit Geraldton.


*Technology is often quite limited, particularly in the earlier meetings - it was often quite difficult to hear what other people were saying in the room. And it was difficult to see who was in the room, so it wasn’t as strong a connection because of technological challenges.*


#### 3.2.4. Assessment against Community of Practice Value Creation Criteria

Communities of Practice operate when a group of people who share a concern or a passion for something they do and learn how to do it better as they interact regularly. Over time, Wenger and colleagues developed a more sophisticated understanding of how communities of practice add value in educational settings. The coding of interview responses against Wenger and colleagues’ framework (Table 1) shows that CoP members derived significant value in its first year, mostly in the first two cycles of the Value Creation Framework of Wenger; the immediate value of forming a community and the potential value for collaborations and projects. 

Members were inspired by the breadth of interest in FDV and the range of discipline areas that engaged in the CoP. Given the complexity of FDV and its damaging and intractable nature, finding colleagues to work with was encouraging for many. One-off cross-disciplinary teaching innovations involving students working directly with FDV issues arose from the new connections made between one member and two others, creating changes in practice (Cycle 3). Limits on the availability of members’ time to devote to the CoP meant these innovations were not able to be taken to the next level of performance improvement (Cycle 4) and Reframing Value (Cycle 5) which would have required deeper and sustained engagement over a much longer time frame.


*Maybe keep it smaller than what it currently is, so it would only be those people who are really passionate and who want to be actively involved. That might be that initially there are the two or three topics we need to address which are, curriculum, being closely connected to the sector and working with the PPI [Public Policy Institute]. So, we have a smaller number of people in designated tasks and activities to keep the momentum going and make sure the action happens rather than talking.*


## 4. Discussion

Universities have an important role in training leaders to address existing and emerging local and global problems. FDV is a complex social issue linked to longstanding inequalities in power and control and where rethinking approaches to prevention and management are needed. This requires truly intersectoral thinking and action. There was considerable enthusiasm among some staff at UWA to consider utilising a Community of Practice approach to initiate efforts to develop more interprofessional ways of thinking and working. Finding colleagues to work with was encouraging for many. We recognised that the CoP would include discussions on the topic of interpersonal and sexual violence which have the potential to be distressing for some people. However, the CoP attracted those people who were particularly aware of the issues of FDV either through their personal or professional experience with many expressing their appreciation of the opportunity to have such conversations about how to move forward in training and responding more effectively to the issues of FDV.

One-off cross-disciplinary teaching innovations involving students working directly with FDV issues arose from the new connections made between some members, creating changes in practice (Cycle 3). Along with broadening involvement to other disciplines, members were keen to continue efforts to have the university acknowledge FDV as a serious issue, arguing it was an important and suitable major challenge that the university could use to showcase the expertise and commitment that existed among staff, enhance cross-disciplinary teaching and research, and inform public policy responses to the complex issues associated with FDV. However, at the time when this evaluation was undertaken, these innovations were not yet taken to the next level of performance improvement (Cycle 4) and Reframing Value (Cycle 5). Such developments would have required deeper and sustained engagement over a longer time frame. By early 2020, the global pandemic COVID-19 had disrupted plans for the year with academics scrambling to reframe their usual content and deliver it online. Travel and placements became impossible for a period, and the demands for developing online learning overwhelmed the potential for further development of novel interdisciplinary learning approaches. Those disruptions with their personal and professional consequences continue [20]. It is possible that more systemic levels of value creation could have progressed had efforts to refine the CoP based on the initial evaluation been possible.

The two refinements to the CoP suggested by many interviewees were that meetings become less regular and that they focus more on the progression of collaborative projects with tangible outputs and sharing what they were doing, with less emphasis on foundational conceptual knowledge. Most participants indicated that although the relationship-building stage and sharing of knowledge was a critical requirement for establishing the CoP, after a year more tangible products should be underway. The products suggested were generally educational material that could engage students, again linking to the idea of providing a holistic educational approach to teaching. Quality educational material could also be utilised by facilities outside of the University of Western Australia, including schools and to support initiatives in rural areas. Any focus on tangible products was recognised as requiring a commitment of time and resources that many interviewees acknowledged that they didn’t possess.

Members of the CoP consistently expressed the view that any future university CoP grant projects should allow funding for a facilitator role as recommended by the literature [6]. Although emerging in response to a small grant funding opportunity that encouraged the establishment of CoPs within the university, the most expensive aspects of the CoP was staff time which was not covered by the CoP funds. Nevertheless, the funding provided was a catalyst and gave the university’s imprimatur for staff to spend time on the endeavour. Despite this, none of the CoP members interviewed spoke of their work with the CoP as contributing to the Scholarship of Teaching and Learning, where efforts to innovate teaching and learning might be acknowledged through the university and be relevant for promotion [21]. In the absence of a Scholarship of Teaching and Learning framework, as one CoP member put it, a commitment to devote time to innovation such as the CoP appears to be a choice to engage in service that will “benefit the university or community” rather than “something that will help your career”.

Challenges of attendance, time commitment, and focus have been recognised by other developing CoPs within university contexts [22,23]. The culture and practice of an institution and the efficacy of its leaders can either encourage and support the innovative practice or constrain it [21]; for innovation to flourish, energetic and creative innovators are needed at all levels of the institution, creating supportive policy as well as directly planning and implementing new approaches. When participation in teaching and learning innovation is recognised and built into individual performance evaluation, universities give value to this scholarship [24,25]. However, without formal recognition and reward, and realistic workloads to accommodate the time required, passion for aiding an improved response to complex social issues is diminished by deadlines, workloads, and other commitments. The timing for implementing this FDV CoP in terms of sustaining it was severely impacted by COVID-19 given its early stage of development. 

Continued engagement with students was seen as a future aspect of the CoP, with a focus on student placement in rural areas such as Geraldton as well as engaging students via possible broadening units, lectures, and other areas of student involvement. A further effort to expand the disciplines involved in the CoP was discussed, involving disciplines such as Education and Social Work (which although included in the original membership had not engaged) and also disciplines such as Business and Commerce. This could create a more integrated, holistic approach to learning and mirror the far-reaching impact of complex issues such as FDV with its impacts across multiple sectors of society. Along with broadening involvement to other disciplines, members were keen to continue efforts to have the university acknowledge FDV as a serious issue through which the university could showcase the expertise and commitment that existed among staff, enhance cross-disciplinary teaching, research, and inform public policy responses to the complex issues associated with FDV.

## 5. Conclusions

Despite its promise of university education that could help create a better future and more rational integrated services for those affected by FDV, this FDV CoP was a casualty of its timing. This was particularly because of COVID-19 and university responses to managing that and other imperatives. It was not able to build on the strong foundation that had been established and carry forward the momentum of its first year. It experienced limitations of time, the lack of a dedicated coordinator, and an inadequate scholarship of teaching and learning focus and senior leadership to reward engagement and embed the CoP within university-wide innovation. Members valued exploring new opportunities for student learning, interdisciplinary perspectives, and establishing relationships with industry partners and across university schools from which future collaborations in research and teaching could emerge. The CoP raised interest within the university in new approaches to deal with complex social problems through cross-disciplinary teaching and FDV as a public policy issue and showed there were academics committed to dealing with the recalcitrant, complex issues related to FDV. However, deeper and sustained engagement over a much longer time frame, along with leadership from the most senior levels of the university would be required for the nascent potential of the first year to be realised. Future efforts should prioritise the creation of tangible products, educational material or teaching approaches, and collaborative research as a key goal for core CoP members.

## Figures and Tables

**Figure 1 ijerph-20-03501-f001:**
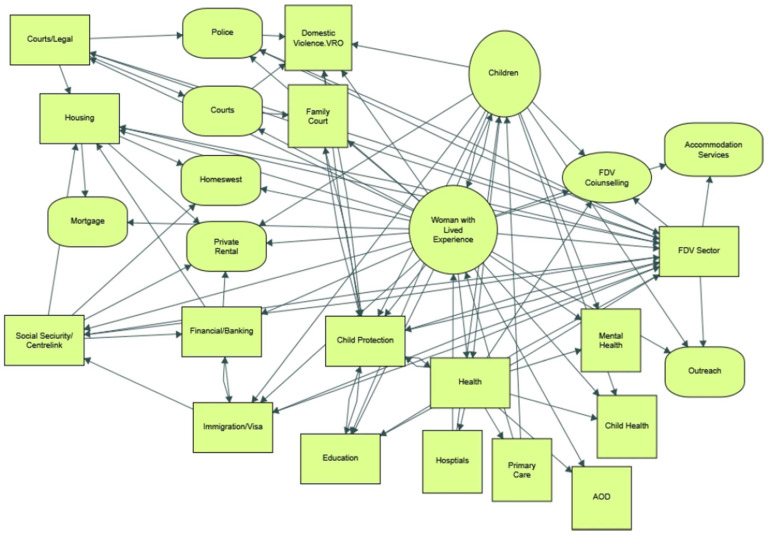
A diagram illustrating the complexity of the needs of a woman experiencing family violence (Source: Courtesy of Professor Colleen Fisher).

**Table 1 ijerph-20-03501-t001:** Cycles of value creation with indicators and indicative responses. Modified from Wenger, Trayner, and de Laat, 2011 [1] (see Appendix A).

Cycle	Indicators of Value	
Cycle 1. Immediate value: indicators of activity/interactions.	This cycle considers networking/community activities and interactions as having value in and of themselves.	“Seeing the will to address this issue, given its complexity.” “Commitment to change the outcome.”
Cycle 2. Potential value: knowledge capital	**Human capital.** New skills, information, perspectives, ideas, as well as confidence, and inspiration **Social capital**-potential opportunities for collaboration and the ability to promote a cause **Resources (tangible capital).** Specific pieces of information, documents, tools and procedures, links and references, search capabilities, visualization tools, and other socio-informational structures that facilitate access to information. **Collective intangible assets (reputational capital).** Reputation of the community or network, the status of a profession, or the recognition of the strategic relevance of the domain. Many people value their community of practice, for instance, for the collective voice or recognition that it provides them in their organization. All these assets increase the potential for collective action. **Transformed ability to learn (learning capital).** The act of participating in a facilitated network or a community as a valuable way of learning can be enlightening for people for whom formal teaching or training methods have always been seen as the only way to learn. When members have experienced significant learning in networks or communities they can transfer this experience to other contexts.	“It’s exposed to me to the operations of the university, it’s been good to understand how the different areas of the university work.” “…it was simply the inspiration in seeing so many different people doing so much different work” “I’ve never been in a room with family law people talking about family law in relation to domestic violence, I’ve always been on the service delivery end, with clients going to family law court. It was interesting to see how the family law bit works.” “I’ve also learnt more personally about how to collaborate with the non-University sector …. when they brought in different organisations …that was that was something I learnt from.” “...a couple of key contacts I made that I feel confident that we will continue work together.” “Sharing resources and ideas, particularly about transformative learning.” “The symposium was a great achievement to show the breadth of work.” “The symposium encouraged everyone to look outwardly from the university to focus their research and work.” “For me, some of the Perth campus-based staff are much more aware of WACRH [in Geraldton], which was definitely one of my aims.” “Symposium was a great event to show what’s happening in the area from different disciplines and external organisations and create connections with them and learn from them.”
Cycle 3. Applied value: Changes in practice	**Indicators of changes in practice.** Knowledge capital is a potential value, which may or may not be put into use. Leveraging capital requires adapting and applying it to a specific situation. e.g. reusing a lesson plan or changing a procedure, implementing an idea, trying a suggestion, enlisting members of one’s network for a cause, or leveraging a collective voice to make a case for an organizational decision.	“UWA pulled together an application to [national organisation] to pilot university-based prevention of FDV – even though it wasn’t successful, it wouldn’t have happened without the CoP.”
Cycle 4. Realized value	**Indicators of performance improvement.** Changes in practice or the use of resources from the community/network that results in improvements in performance of what matters to stakeholders, including members who apply a new practice.	“Projects involving students [Media, WACRH and McCusker] was a good achievement. They were innovations.” “Some of us got involved in a World Universities Network project on FDV to do collaborative international research.”
Cycle 5. Reframing value:	Indicators for assessing the redefinition of success. Social learning causes a reconsideration of the learning imperatives and the criteria by which success is defined. This includes reframing strategies, goals, as well as values.	(no example)

## Data Availability

Data sharing is not applicable to this article due to privacy considerations.

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
