# Peer review of "Addressing Complex Social Problems Using the Lens of Family Violence: Valuable Learning from the First Year of an Interdisciplinary Community of Practice"

_ijerph, 2023, doi:10.3390/ijerph20043501_

Round 1

Reviewer 1 Report

Thank you for the opportunity to review this manuscript. I found it very insightful as it described a new way of learning and exploring solutions to very complex social issues.

Line 85 page 2, delete "toto" to just "to"

I did not see a diagram and I believe that was mentioned, it would be nice to have a graphic depicting the Community of Practice so that we can visually see the model as well as the circles of participants.

Very important topic that I am glad to have reviewed. Family violence is a worldwide problem that deserves this emphasis in science as we explore solutions.

Author Response

Thank you for the opportunity to review this manuscript. I found it very insightful as it described a new way of learning and exploring solutions to very complex social issues.

Thank you

Line 85 page 2, delete "toto" to just "to"

Done as requested

I did not see a diagram and I believe that was mentioned, it would be nice to have a graphic depicting the Community of Practice so that we can visually see the model as well as the circles of participants.

This was uploaded as a Supplementary file as per the journal instructions for Figures. We have now embedded it in the manuscript

Very important topic that I am glad to have reviewed. Family violence is a worldwide problem that deserves this emphasis in science as we explore solutions.

Thank you

Reviewer 2 Report

Interesting paper, shows how it is difficult to carry on interdisciplinary work at University.

I would have liked a little more discussion about the obstacles (if any) in tackling specifically the issue of violence. Is it a subject "easy" to discuss or not?

I also have some problems with the concept of "family and domestic violence", as it lacks a gender dimension. It seems to me an outdated concept/paradigm. In the text, in some occasion it is clear that it is instead "partner violence against women and children". I would like a little more thought and discussion on this issue. I know that it can be controversial, but this should also be discussed!

Some editing problems:

has aa particularly

tion toto a lesser extent toto ?

University the invited staff. A word is lacking ?

Medical Sciences found that 

Medical Sciences and found that ?

priorities)).

FDV the recalcitrant, complex

FDV, and the recalcitrant, complex ?

Author Response

Interesting paper, shows how it is difficult to carry on interdisciplinary work at University.

I would have liked a little more discussion about the obstacles (if any) in tackling specifically the issue of violence. Is it a subject "easy" to discuss or not?

We have added an additional comment in the discussion

“Of course, any CoP includes discussions on the topic of interpersonal and sexual violence which have the potential to be distressing for some people. However, the CoP attracted those people who were particularly aware of the issues of FDV either through their personal or professional experience with many expressing appreciation of the opportunity to have such conversations about how to move forward in training and responding more effectively to the issues of FDV.” 

I also have some problems with the concept of "family and domestic violence", as it lacks a gender dimension. It seems to me an outdated concept/paradigm. In the text, in some occasion it is clear that it is instead "partner violence against women and children". I would like a little more thought and discussion on this issue. I know that it can be controversial, but this should also be discussed!

Thank you for the comment. The reviewer is correct in that different terminologies are used, and in Australia, the national response refers to Violence Against Women and their Children. However, in Western Australia, the state government uses the term Family and Domestic Violence in their 10-year strategy and so that was the term we adopted for the CoP, and family violence was the term generally utilised within the efforts of the Community Respect Equality (CRE) Action plan in Geraldton which is referred to in this paper. We have now referred to issues related to terminology and the gendered nature of FDV in the introduction (lines 153-166)

Some editing problems:

has aa particularly   (corrected)

 toto a lesser extent toto ?  (corrected)

 University the invited staff. A word is lacking ?  (the deleted)

Medical Sciences found that 

Medical Sciences and found that ? (and added)

 priorities)).  (bracket deleted)

FDV the recalcitrant, complex

FDV, and the recalcitrant, complex ?  (superfluous FDV deleted)

 Thank you for identifying these. They have now all been corrected.

Reviewer 3 Report

I have read the proposed manuscript with great interest. I fully agree with the authors that the problem of domestic violence in all its forms receives little attention in the university. Based on this consideration, I commend the authors for addressing this issue. The paper is well written and enjoyable to read, but I think the clarity and immediate understanding would benefit if the authors presented the findings schematically with a picture. In this regard, I note that Figure 1 is not included in the text.

Further comments:

- how come the authors did not include sexual abuse in the introduction (lines 150-151)? In general, sexual abuse and gender-based violence in the family are completely absent. Is there a particular reason for this decision?

- in the methods (line 228), the authors state that they interviewed 12 members who attended at least half of the 8 meetings. It would be interesting to get more information about these surveyed members, even if it is just their area of expertise, to better contextualize their responses in the results section.

Overall, it seems that the proposed results do not add much to what is already known or predictable in the literature. This is perhaps the major limitation of the article. However, the authors' discussions and reflections are quite interesting and appropriate.

Author Response

I have read the proposed manuscript with great interest. I fully agree with the authors that the problem of domestic violence in all its forms receives little attention in the university. Based on this consideration, I commend the authors for addressing this issue. The paper is well written and enjoyable to read, but I think the clarity and immediate understanding would benefit if the authors presented the findings schematically with a picture. In this regard, I note that Figure 1 is not included in the text.

Figure 1 was uploaded as a Supplementary file as per the journal instructions for Figures. We have now embedded it in the manuscript.  It well describes the complexity of FDV responses and addresses what the reviewer requested.

Further comments:

- how come the authors did not include sexual abuse in the introduction (lines 150-151)? In general, sexual abuse and gender-based violence in the family are completely absent. Is there a particular reason for this decision?

We have now elaborated relevant issues in the paragraph.

Lines 151-155  In the CoP, we used the term Family and Domestic Violence although Australia’s national policy response explicitly identifies the gendered nature of this issue and uses the term violence against women and their children. However, in Western Australia, the state government uses the term Family and Domestic Violence in their 10-year strategy and so that was the term we adopted for the CoP, and in this paper.

Lines 166-170 Within the CoP, the gendered nature of FDV was discussed and understood. We did not focus on sexual abuse as often it is physical violence that elicits a victim reporting or law and justice responses, and we were keen to broaden awareness of the pernicious nature of coercive control which is under recognised, more difficult to deal with and where public responses are under developed.

- in the methods (line 228), the authors state that they interviewed 12 members who attended at least half of the 8 meetings. It would be interesting to get more information about these surveyed members, even if it is just their area of expertise, to better contextualize their responses in the results section.

We have added more information to the paper regarding the participants but we are need to be non-identifying so the information on their backgrounds under represents their breadth of experience.

“The evaluation was undertaken with the most regular attendees at the CoP meetings and represented those based in law (3), health, albeit from very different disciplinary back-grounds including public health /health promotion, medicine, rural health, research and social sciences, (8), and media and communications (1)”.

Overall, it seems that the proposed results do not add much to what is already known or predictable in the literature. This is perhaps the major limitation of the article. However, the authors' discussions and reflections are quite interesting and appropriate.

We appreciate the positive comments and are encouraged that the reviewers overall recognise the importance of the issue and that universities need to do more in this space. We hope this publication will help others interested in creating the future we hope for in terms of enhancing women’s health and wellbeing.
